# Cancer Stemness/Epithelial–Mesenchymal Transition Axis Influences Metastasis and Castration Resistance in Prostate Cancer: Potential Therapeutic Target

**DOI:** 10.3390/ijms232314917

**Published:** 2022-11-29

**Authors:** Enrique A. Castellón, Sebastián Indo, Héctor R. Contreras

**Affiliations:** Laboratory of Cellular and Molecular Oncology, Department of Basic and Clinical Oncology, Faculty of Medicine, University of Chile, Santiago 8380453, Chile

**Keywords:** prostate cancer, Epithelial–Mesenchymal transition, cancer stem cell, castration resistance, metastasis

## Abstract

Prostate cancer (PCa) is a leading cause of cancer death in men, worldwide. Mortality is highly related to metastasis and hormone resistance, but the molecular underlying mechanisms are poorly understood. We have studied the presence and role of cancer stem cells (CSCs) and the Epithelial–Mesenchymal transition (EMT) in PCa, using both in vitro and in vivo models, thereby providing evidence that the stemness–mesenchymal axis seems to be a critical process related to relapse, metastasis and resistance. These are complex and related processes that involve a cooperative action of different cancer cell subpopulations, in which CSCs and mesenchymal cancer cells (MCCs) would be responsible for invading, colonizing pre-metastatic niches, initiating metastasis and an evading treatments response. Manipulating the stemness–EMT axis genes on the androgen receptor (AR) may shed some light on the effect of this axis on metastasis and castration resistance in PCa. It is suggested that the EMT gene SNAI2/Slug up regulates the stemness gene Sox2, and vice versa, inducing AR expression, promoting metastasis and castration resistance. This approach will provide new sight about the role of the stemness–mesenchymal axis in the metastasis and resistance mechanisms in PCa and their potential control, contributing to develop new therapeutic strategies for patients with metastatic and castration-resistant PCa.

## 1. Introduction

Prostate cancer (PCa) is a leading cause of oncologic death in men, worldwide [1]. Although new screening programs have increased the speed of early diagnosis and timely treatments with curative intentions [2], the high rate of relapse and metastasis remains as the major challenges [3,4]. At the beginning, PCa is sensitive to androgen action [5]. Testosterone, and its prostate metabolite dihydrotestosterone (DHT), induces cell proliferation, tumor growth, and probably, its dissemination [6]. For this reason, the treatments involving androgen deprivation therapy (ADT), when curative surgical prostate resection is not possible, have being developed [7,8]. Pharmacological castration using GnRH analogs, in order to block the hypothalamus–pituitary–testicular axis, provides the first-line therapy for disseminated PCa [7]. Nevertheless, at certain point of the treatment, the cells become androgen insensitive, resulting in a castration-resistant PCa (CRPC) with a poor prognosis [9]. The genetic and molecular mechanisms of androgen resistance are complex, and they are not completely understood [10]. The evidence suggests that, in some cases, the androgen receptor (AR) is involved in this resistance [8,11]. Gene amplification, mutations and other alterations in the AR gene have been reported [10]. In recent years, many articles have been published about AR-variant 7 (AR-7, a constitutively activated AR receptor), which is increased in CRPC, and it has been proposed as a prognostic biomarker, and it has been described as having an ligand-independent activated form [12,13,14]. In addition, alterations in androgen metabolism, and the local biosynthetic pathway within the prostate gland have been associated with androgen sensitivity [15,16]. Probably, CRPC is the result of a combination of these different mechanisms. On the other hand, recurrence and metastasis progression are also complex processes, involving a variety of mechanisms and genomic alterations of malignant cells [17,18,19]. It is well known that the epithelial–mesenchymal transition (EMT) is the main pathway by which the malignant epithelial cells (from carcinomas) switch their genetic program toward a mesenchymal phenotype, acquiring the characteristic hallmarks of cancer cells, such as invasive and metastatic behaviors, among others [18,20,21,22,23,24]. As a result, an epithelial cell loses its polarity, proliferation, differentiation and positioning controls, changing into a mesenchymal phenotype [25]. Interestingly, we have reported that ZEB1, a key EMT factor, is involved in the regulation of androgen synthesis in PCa cells [26]. However, increasing evidence indicates that tumors contain a heterogeneous cell population, [27] and probably, a cooperative action of these different types of malignant cells is needed to accomplish a successful metastatic process. In the last decade, a small subpopulation of malignant cells with stemness features have been identified and characterized in many cancers including PCa [4,28,29,30]. This population, called cancer stem cells (CSCs), has been proposed as responsible for relapse and metastasis [4]. In recent years, our group has contributed to this field in PCa [31,32,33].

## 2. Epithelial–Mesenchymal Transition and Cancer Stem Cells in Prostate Cancer

EMT occurs normally throughout embryonic development. A primary EMT occurs early in embryonic development (before implantation), and it continues after implantation during the mesoderm formation. Then, a secondary EMT takes place during mesodermal cells division after gastrulation. Finally, a tertiary EMT occurs during the organ formation stage [34]. During carcinogenesis (in carcinomas), a similar process takes place that transforms a malignant epithelial cell into a highly invasive mesenchymal-like cell. This process has been also called EMT [35,36,37]. The epithelial malignant cells progressively lose adhesion molecules such as E-cadherin, syndecans and tight junction molecules, while the gene expression factors such as SNAIL1, SNAI2/Slug and TWIST increase their expression together with the mesenchymal markers such as vimentin, N-cadherin and metalloproteinases, resulting in a migrating and invasive phenotype [38]. On the other hand, we have found that ZEB1 represses Syndecan 1 expression and promotes an aggressive phenotype in PCa cells. A detailed study of this process in PCa has been carried out in our laboratory [39,40,41,42,43]. There is evidence that this mesenchymal and invasive cell phenotype is involved in the metastatic process 19 [23]. Nevertheless, there is no direct proof that these mesenchymal cells have also a colonizing capacity. On the other hand, there is growing evidence suggesting that CSCs, which are present in most tumors as a small population of malignant cells, are finally responsible for relapse and metastasis [4,44,45]. In addition, we have identified and studied a CSCs population obtained from samples of human PCa. We have determined the molecular signature of stemness (CD133+/CD44+/ABCG2+/CD24-) [31] of this population, and we have evaluated its proliferation, migration, invasion and clonogenic capacity [32]. We were able to separate this CSCs population from the mesenchymal cancer cells (MCCs) by modifying the cultured conditions, which was followed by magnetic-associated cells sorting (MACS) [31]. In adherent conditions, most of the cells remain in mesenchymal-like state as they are evaluated by specific markers and functional assays. However, in non-adherent conditions, most of the mesenchymal adherent cells die by anoikis (anchorage-dependent apoptosis), and a few cells survive and rapidly form spheres that grow and remain for several weeks. After the MACS separation, the sphere-forming cells represent the enriched CSCs population. These CSCs were characterized by a functional assay, presenting a lower proliferation rate, an increased resistance to apoptosis and drugs treatments, a reduced invasive properties, and a high clonogenic capacity compared with the MCCs (Figure 1). In addition, these CSCs show no expression of GnRH-R or AR as well as many differentiation markers [32]. On the other hand, we have produced CSCs that were knocked down for several stemness genes such as Sox2, Klf4 and Myc. In these conditions, the cells reverse their phenotype toward a more mesenchymal form. Additionally, silencing Sox2 in the CSCs resulted in no metastatic progression in an orthotopic murine model (unpublished results). In addition, we isolated and characterized the miRNAs from CSCs exosomes and evaluated their possible association with metastasis [33].

## 3. Cancer Stem Cells and Epithelial–Mesenchymal Transition Control Prostate Cancer Progression

The CSCs represent less than 1% of a primary tumor even though have been suggested to drive tumor progression, relapse and metastasis [4,46,47]. The CSCs can respond to stressing conditions (hypoxia, oxidative damage, xenobiotics, etc.), modifying the gene expression program toward a migrating, invasive and resistant phenotype. It is believed that the CSCs may originate from transformed epithelial cells through EMT, leading to a stemness gene expression program [48]. On the other hand, to generate migrating and invading cells, the CSCs have to undergo EMT, and the EMT marker expression represents a PCa progression indicator. The EMT predictors include an increase in the N-cadherin and vimentin expression and a decrease in E-cadherin, EpCAM, and other epithelial markers [38]. When the migrating and invading cells come out from the primary tumor and reach the blood stream, they are usually called circulating tumor cells (CTCs). A small population of these CTCs survives in circulation, colonize distant tissues, proliferate and originate metastasis [49]. EpCAM is generally used to identify cancer cells in the circulation of PCa patients. Systems to detect the CTCs are based on EpCAM positivity, and several studies indicate that the number of CTCs-EpCAM+ increased with the PCa progression. However, if the stemness phenotype is responsible for metastasis, it is probably that neither EpCAM nor E-cadherin would be expressed on the CSCs [50,51]. When we were comparing normal men, localized PCa and metastatic patients, a higher number of CTCs-EpCAM+ was found in the last group [52]. On the other hand, in a transgenic mice model, only the cells undergoing EMT were capable of self-renewal compared with their epithelial and mesenchymal counterparts [4,53]. To add more complexity to the functional relationship between stemness and EMT, when E-cadherin was silenced in spheres that were obtained from a PC3 human PCa cell line, the EMT process was stimulated [54], while the E-cadherin expression induced stemness gene expression and sphere formation in DU 145 PCa cells [55,56]. These results indicated that a further investigation is needed to understand the stemness–EMT axis and its influence in metastasis and resistance. It is highly probably that both the CSCs and MCCs are also heterogeneous sub-populations. Among several other progression and metastasis markers in PCa, CD117 (c-Kit receptor) has been found to be highly expressed in PCa patients with high-grade tumors in comparison with those with low-grade tumors. CD133, one of the more well-known stemness markers, is increased in high-Gleason tumors, but it is not present in CTCs [57]. CD44, another stemness marker, was found to be expressed in invasive and self-renewing PCa cell lines together with other EMT and stemness markers [58,59]. In addition, the CD44 stemness marker was found to be expressed in patients who tested positive for chromogranin A [60], a neuroendocrine cell marker which is present in a very aggressive and resistance form of PCa. This suggests that PCa neuroendocrine cells (negative for AR and PSA) may be associated with CSCs, or they represent a subpopulation of them [61,62]. PCa neuroendocrine cells also express stemness markers, and identifying the relationship between these cell populations would shed light on the castration resistance progression of PCa, especially the neuroendocrine type. Recently, a cross-regulation of key EMT (SNAI2/Slug) and stemness (Sox2) genes have been reported [63,64,65], supporting the hypothesis that a stemness–EMT axis is operating and providing the necessary plasticity to guarantee the maintenance of tumor heterogeneity [66]. Recently, Zhao et al., have reported that Slug promotes hepatocellular cancer cell progression by increasing Sox2, but this study was performed in cell lines and xenografts in nude mice for a tumor growth evaluation [67]. Additionally, there is recent evidence that Sox2 can regulate AR and lineage plasticity in PCa cell lines and a xenograft model [68]. Moreover, there is increasing evidence, in several cancers, that this stemness–EMT axis is operating to promote cancer progression [69,70,71,72,73].

## 4. Role of Cancer Stem Cells in Metastatic Colonization and Progression

It has been calculated than more than 3 million cells per gram of tissue come out from the tumor to circulation every day in an average cancer patient. However, less than <0.01% of these cells can originate from a clinical metastasis [4,49,74]. This means that this lethal process is highly inefficient, even though it kills most of the cancer patients [75]. Once they are in the blood stream, the cancer cells from the tumor are called circulating tumor cells (CTCs). The subpopulation of CTCs that are able to colonize metastatic niche is often called metastasis-initiating cells (MICs), and when these cells grow in the colonized niche, they are named disseminated tumor cells (DTCs). Each cell kind is a sub-population of the predecessor. There is a consensus to call the CSCs to those cells that are able to colonize and develop micro-, and then, macro-metastases [76,77]. On the other hand, the capacity to leave the tumor, come into circulation, survive in circulation, colonize the selective tissue, survive in the metastatic niche, and finally, grow in this niche, requires EMT and the plasticity of the CSCs [66,75,78]. It is evident then, that not all CTCs, even not all DTCs, are able to develop metastasis. Many CTCs (rather, most of them) die in circulation. Even many DTCs can remain quiescent or die in the metastatic niche due to adverse microenvironment conditions. Therefore, only a small subpopulation (CSCs) has the ability to grow in the metastatic niche, and this recapitulates the primary tumor heterogeneity in a secondary site [76,79]. The stemness features of the CSCs, as asymmetric division and pluripotency, allows them to maintain the CSCs population as well as cell differentiation (EMT) to give rise all of the cell varieties of the original heterogeneous tumor [80]. In PCa, an increasing number of specific markers have been associated with tumor progression and therapeutic resistance. These markers have been found in CTCs and in bone metastasis DTCs. CXCR4 (SDF-1 chemokine receptor), EpCAM and EZH2 were found to be associated with relapse and distant metastasis, and clinical studies [81,82] suggest that these markers might drive metastasis. Given that some markers were restricted only in the prostate tumor suggests that they probably are not directly involved in the niche colonization and metastasis progression. However, CD117 and CXCR4 were expressed in bone metastatic foci at the levels shown in the primary tumor [4,83,84,85]. These findings suggest that CD117 and CXCR4 may be implicated in driving colonization, metastasis progression and dormancy escape. In addition, E-cadherin expression has been found to be associated with bone metastasis in clinical studies, giving more evidence that a mesenchymal–epithelial transition (an inverse EMT process) might be occurring [86]. Even when the specific molecular mechanisms for niche colonization survival, dormancy escape and further metastasis progression in the bones of PCa patients are still a challenge to be fully addressed, the evidence accumulated so far indicates that the CSCs play a crucial role.

## 5. Different Malignant Cell Types Collaborate to Produce Distant Metastasis in Prostate Cancer

Revisiting the role of CSCs in metastasis, the fact that these cells with little invasive activity can leave the tumor and colonize the pre-metastatic niches strongly suggests that some kind of collaboration with highly invasive MCCs is occurring. Recently, Dr. Thomson’s group provided the first evidence about this potential cooperative action. Using commercial cell lines derived from PCa (PC3), they enriched a cell population with metastatic cells (TICs) using a strong epithelial program. In turn, they reduced the cell population with TICs using a mesenchymal program. The over-expression of mesenchymal genes in the first population decreased their TIC ability, whereas the knocking down of these genes enhanced the TIC capacity in the second population. Using immunocompromised NOD/SCID mice, they observed that when they were injected in combination, the mesenchymal cells increased the metastatic potential of the epithelial TIC-enriched population, suggesting a cooperative action between both of the cell types [87]. This hypothesis supports the idea that within a tumor, through EMT, the mesenchymal cells became the predominant population, giving the tumor a fully invasive capacity. However, it can be proposed that a small cell population that expresses a stemness cell program (CSCs) remains in the tumor, and this can escape passively with the bulk of the MCCs. Once they are in the metastatic niche, the CSCs can proliferate and produce progenitor cells that may further differentiate to an epithelial-like phenotype. This may explain our results (unpublished) and other observations, showing that in metastatic PCa samples, an increase in the epithelial markers and a decrease in the mesenchymal markers are observed, which has been called mesenchymal–epithelial transition (MET) [88,89]. Probably, the metastatic foci will generate the full heterogeneity of the original tumor, in which the epithelial-like cells will go through EMT again and probably keep a few CSCs. This is an interesting hypothesis that is worth proving using the CSCs and MCCs derived from patient’s tumors. In our laboratory, we have developed an orthotopic murine model for human PCa using NOD/SCID mice [90]. In this model, we have injected both CSCs and MCCs obtained from patient explants. Both of the types of cells produced metastasis. However, the prostate tumor from the CSCs was smaller than those from the MCCs. In addition, the metastasis from the CSCs was obtained more slowly that with the MCCs (unpublished). We interpret that the CSCs take longer to generate progenitors and all of the heterogeneity of the MCCs through EMT, and rather, the MCCs are already a mixed population including migrating and invading cells. These cells, through EMT, produce a population of CSCs that provide a metastatic capacity. This process seems to be faster than the other one. Taken together, these results and the others reported in different cancer types, such as hepatocellular carcinoma, strongly support the stemness–EMT axis, in which the driver genes for two processes are cross-regulated. This defines an epithelial-to-mesenchymal plasticity of the CSCs [66]. This axis may turn the focus of interest to identify new therapeutic targets.

## 6. Cancer Stems Cells and Epithelial–Mesenchymal Transition in Relapse and Castration Resistance

For the locally advanced PCa, radical prostatectomy and radiotherapy are the main treatments. When the PCa is disseminated, ADT and chemotherapy are the options [2,5,7]. Other treatments such as immunotherapy (i.e., dendritic cell-based vaccine Sipuleucel-T) have been developed recently [91,92]. For locally advanced PCa, the treatments have a curative intention. However, local recurrence and metastasis (and following castration resistance) are the main concerns after surgery or radiation (>30%). It is highly probably that the CSCs that remain in the surgical niche, in the irradiated tissue or in the circulation can promote the development of local relapse and/or distant metastasis. This is explained by the fact that the CSCs have been found to be resistant to most therapies, causing DNA damage in highly proliferative cells or directed to hormonal/signaling targets that affect mainly the bulk tumor cells but not the CSCs [93]. The CSCs markers such as CD117, EZH2, among others, which are expressed in the prostate primary tumor, resulted in being predictive for biochemical recurrence (rising PSA levels in the circulation) after radical prostatectomy [57,94]. It has been found that number of CTCs-CD117+ were elevated for more than 3 months after the radical prostatectomy in patients that later underwent biochemical recurrence [56], suggesting that circulating CSCs may be a useful predictor for early relapse [95]. Other clinical studies have shown that CRPC or the neuroendocrine PCa type have different CTCs (AR-low) [4,96]. In the murine models, CD166, a recently proposed stemness marker for colorectal cancer, was found to be up-regulated in PCa after castration [97]. In humans, EZH2 was increased in advanced PCa, and it was associated with poor survival [98]. All of this evidence points out that the CSCs have a major role in relapse and metastasis in PCa. Moreover, we and other groups have found that the CSCs show a significant resistant to chemotherapeutics and radiation [32,99]. In other studies, it was reported that the inhibition of CXRC4 significantly increased the PCa cell lines to docetaxel, which is one of the most common chemotherapeutic drugs used in PCa [100]. Additionally, silencing the EpCAM gene in PCa cell lines increases the patient’s sensitivity to radiation and chemotherapeutics in vivo [101]. Furthermore, other CSCs markers have been studied regarding therapeutic resistance. We have analyzed the influence of ATP-binding cassette (ABC) transporters, including ABCG2 (specific stemness marker), in PCa resistance, and we have found that several ABC transporters are over-expressed in the PCa cells. The pharmacological inhibition or knocking down of these ABC pumps resulted in a significant increase in drug sensitivity [102,103]. Our group and others have found that the ABCG2 transporter is highly expressed in the spheres from PCa which correlates with a level of high drug resistance of this CSCs population [32,104]. In several other studies, the CSCs have showed to have a level of high resistance to multiple drugs including taxanes, tyrosine kinase and topoisomerase inhibitors, among others [4,105,106,107]. ALDH1, which is another stemness marker, has been implicated in changes in the metabolism of chemotherapeutic agents, reducing radio-sensitivity in the PCa cell lines [105]. In patients treated with enzalutamide or abiraterone, the CTCs showed an increased expression of the AR-7 splicing variant and a decreased progression-free and overall survival [108]. In addition, AR-7 has been found to be expressed in the CSCs undergoing tumor progression during ADT [109,110]. Interestingly, the evidence indicates that AR expression can be induced in the CSCs during the progression to castrate resistance [111]. Additionally, the EMT marker SNAI2/slug is encoded by an androgen-regulated gene [112]. In addition, SNAI2/Slug increases the AR expression and binds to it, acting as co-activator and increasing the AR activity regardless of the androgen being absence. This has been proposed as a mechanism driving castration resistance [112]. Considering that SNAI2/Slug also cross-regulate with Sox2, and Sox2 can regulates AR [68], it is plausible that the stemness–EMT axis has a major role in castration resistance in PCa.

## 7. Orthotopic Model for the Study of Human Prostate Cancer Metastasis

As stated above, an NOD/SCID mouse has been widely used for the metastasis study of several human cancers [113]. A critical issue is the route by which the human cancer cells are injected. Many authors use subcutaneous, intravenous, or intra-cardiac administrations with different results. Lastly, orthotopic models has been developed (injection in the same mouse organ or tissue from which human cell derives). This model mimics, more accurately, the metastatic process. Lastly, a few reports on the orthotopic models for human PCa have been published [114,115]. We have developed a modification of an orthotopic model for PCa using a cell injection in one of the anterior lobes of the NOD/SCID mouse prostate [90]. This orthotopic injection resulted in a consistent and reproducible metastatic progression. Firstly, a fraction of tumor cells injected in the mouse prostate survived and generated a tumor derived from the injected cells (transduced with luciferase and red fluorescent protein genes). In a chronological sequence, the metastatic foci begin to appear in the liver, lungs and kidneys. The injection of the cell into the anterior lobe, instead the ventral prostate, has the advantage that it is possible to surgically remove the prostate tumor in order to evaluate the progression of the metastasis with or without the main “primary” prostate tumor. In this model, we have demonstrated the utility of prostatectomy during metastasis progression [116], and the effect of knocking down the stemness gene Sox2 on driving metastasis. In the current studies, we are establishing the progression toward a CRCP using surgical castration as an ADT. We consider that this orthotopic pre-clinical model represents a very suitable system to further study of relapse, resistance and metastasis of PCa.

## 8. Concluding Remarks

It may be suggested that the EMT gene SNAI2/Slug up regulates the stemness gene Sox2, and vice versa, inducing an androgen receptor expression, promoting metastasis and castration resistance in prostate cancer. This hypothesis is based on recent separate information about the influence on the CSCs and the EMT process in metastasis, relapse and treatment resistance in many cancers, including PCa. Recent evidence indicates that the generation of CSCs is dependent on EMT. It has been shown that several EMT factors increase the number of pluripotency genes. One of the best candidates is the SNAI2/Slug transcription factor. On the other hand, several stemness genes have been identified in PCa as being one of the most important, e.g., Sox2. This stemness transcription factor also regulates the EMT markers, establishing a stemness–EMT axis that allows it to generate the cell tumor heterogeneity. Additionally, Sox2 regulates several other differentiation genes such as the androgen receptor. The CSCs can originate from the cells undergoing EMT, and conversely, they are also capable of generating mesenchymal cells through differentiation. Both the stemness and EMT genes may inter-regulate their transcriptions. On the other hand, CRPC (the most lethal form of this cancer) has been also associated with the stemness–EMT axis. Takin the evidence together, it can be proposed that this stemness/EMT axis may promote androgen sensitivity changes, conducting to a castration resistance condition and metastasis (Figure 2).

Based on the background that is discussed above, it would be valuable to propose the study of the effects of manipulating the stemness–EMT axis genes SNAI2/Slug and Sox2, among others, on metastasis and castration resistance in PCa. The potential results of such a study would contribute to the understanding of the role of CSCs and EMT process upon the genetic, cellular and molecular mechanism of metastasis and hormone-resistance in PCa because these are still the major hints and challenges of the high mortality of this disease. Furthermore, new insight on these aspects obtained from pre-clinical models will have an impact on identifying new therapeutic targets for clinical use.

## Figures and Tables

**Figure 1 ijms-23-14917-f001:**
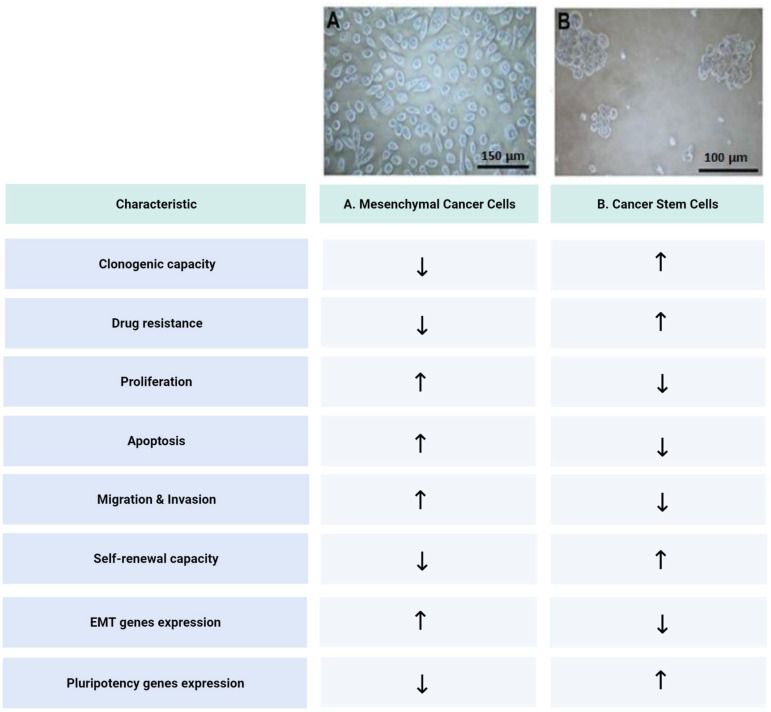
Comparative summary of the main characteristics of mesenchymal cells and cancer stem cell from prostate cancer. (**A**) Mesenchymal cancer cells; (**B**) Cancer stem cells. ↑: Increased. ↓: Decreased.

**Figure 2 ijms-23-14917-f002:**
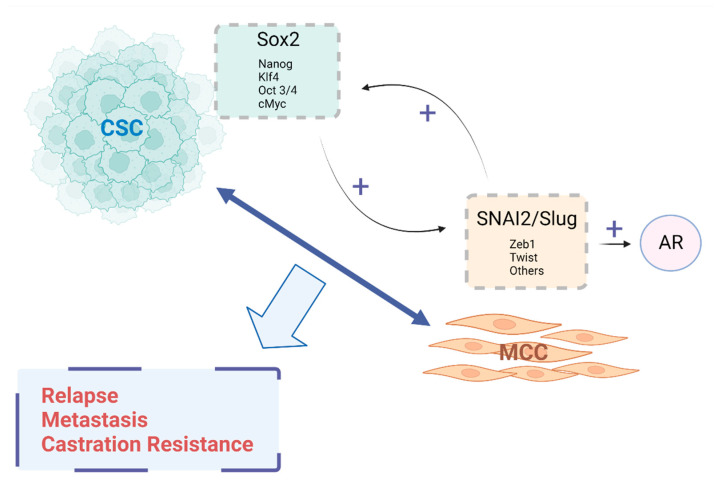
Proposed model for stemness–EMT axis in prostate cancer. CSC: cancer stem-like cell; MCC: Mesenchymal-like cancer cell; AR: Androgen Receptor.

## Data Availability

Not applicable.

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
