# Peer review of "Cancer Stemness/Epithelial–Mesenchymal Transition Axis Influences Metastasis and Castration Resistance in Prostate Cancer: Potential Therapeutic Target"

_ijms, 2022, doi:10.3390/ijms232314917_

Round 1
Reviewer 1 Report
Manuscript is well-organized and clearly presented. The references are properly chosen and up to date. The idea of the study is interesting and worth to be presented. In general I don't have any confusions or suggestions. I may only feel small insufficiency in the number of figures/graphs. This one at the very end of the paper clarifies in nice manner the idea, but in previous paragraphes there is lack of it. It would improve value.
Author Response
Reviewer 1:
Manuscript is well-organized and clearly presented. The references are properly chosen and up to date. The idea of the study is interesting and worth to be presented. In general I don't have any confusions or suggestions. I may only feel small insufficiency in the number of figures/graphs. This one at the very end of the paper clarifies in nice manner the idea, but in previous paragraphes there is lack of it. It would improve value.
Answer: We appreciate the reviewer comments. We have taken in consideration the suggestion and have included an extra figure in the revised manuscript.
Reviewer 2 Report
In the manuscript, the authors describe the role of cancer stem cells (CSCs) and epithelial-mesenchymal transition (EMT), and the vision of EMT-related genes to upregulate stemness genes in the treatment of prostate cancer. However, there still are some issues in the manuscript that require further discussion.
1. The review of 8 parts of your manuscript has some unique insights, but can you make some sequential changes in the logical relationship, for example, does the logical line order of parts 4,5, and 6 need to be adjusted? At the same time, we can conclude from the title, of part 2 "Epithelial-mesenchymal transition and cancer stem cells in prostate cancer" and 6 "Cancer stems cells and epithelial-mesenchymal transition in relapse and castration resistance" logic some repeated?
2. The author reviewed the definition of EMT in the second part, but the previous EMT related content has been mentioned many times, can it be explained in advance in the previous part?
3. There are some concepts that still need to be discussed mentioned in part 2”EMT occurs normally throughout embryonic development” Does EMT exist during throughout embryonic development or during early embryonic development?
4. In the conclusion section, the authors should set a follow-up discussion.
Author Response
Reviewer 2:
In the manuscript, the authors describe the role of cancer stem cells (CSCs) and epithelial-mesenchymal transition (EMT), and the vision of EMT-related genes to upregulate stemness genes in the treatment of prostate cancer. However, there still are some issues in the manuscript that require further discussion.
- The review of 8 parts of your manuscript has some unique insights, but can you make some sequential changes in the logical relationship, for example, does the logical line order of parts 4,5, and 6 need to be adjusted? At the same time, we can conclude from the title, of part 2 "Epithelial-mesenchymal transition and cancer stem cells in prostate cancer" and 6 "Cancer stems cells and epithelial-mesenchymal transition in relapse and castration resistance" logic some repeated?
Answer: We appreciate the comments of the reviewer. For more logical sequence we have change the order of some parts of the revised version of the review according to the suggestion.
- The author reviewed the definition of EMT in the second part, but the previous EMT related content has been mentioned many times, can it be explained in advance in the previous part?
Answer: We agree with the reviewer comment and have included the EMT definition in the introduction to avoid repetitions.
- There are some concepts that still need to be discussed mentioned in part 2”EMT occurs normally throughout embryonic development” Does EMT exist during throughout embryonic development or during early embryonic development?
Answer: We thank the reviewer comment. Actually, EMT takes place during most embryonic development. A called primary EMT occurs early in embryonic development (before implantation), and continues after implantation during the formation of the mesoderm from the primitive ectoderm (gastrulation). Then, a secondary EMT takes place as a differentiation process that produces mesenchymal cells with restricted ability to differentiate (mesodermal cells division after gastrulation into different parts of mesoderm). Finally, a tertiary EMT occurs during organs formation. We have specified briefly this point in the revised manuscript.
- In the conclusion section, the authors should set a follow-up discussion.
Answer: We agree with the reviewer and have included an additional paragraph in the revised conclusion section.